# Development of a longevous two-species biophotovoltaics with constrained electron flow

Huawei Zhu [1,2], Hengkai Meng[1,3], Wei Zhang[1,3], Haichun Gao[4], Jie Zhou[1], Yanping Zhang[1] & Yin Li[1]

Microbial biophotovoltaics (BPV) offers a biological solution for renewable energy production by using photosynthetic microorganisms as light absorbers. Although abiotic engineering approaches, e.g., electrode modification and device optimization, can enhance the electro-chemical communication between living cells and electrodes, the power densities of BPV are still low due to the weak exoelectrogenic activity of photosynthetic microorganisms. Here, we develop a BPV based on a D-lactate mediated microbial consortium consisting of photo-synthetic cyanobacteria and exoelectrogenic *Shewanella*. By directing solar energy from photons to D-lactate, then to electricity, this BPV generates a power density of over $150\ mW\cdot m^{-2}$ in a temporal separation setup. Furthermore, a spatial-temporal separation setup with medium replenishment enables stable operation for over 40 days with an average power density of $135\ mW\cdot m^{-2}$. These results demonstrate the electron flow constrained microbial consortium can facilitate electron export from photosynthetic cells and achieve an efficient and durable power output.

[1] CAS Key Laboratory of Microbial Physiological and Metabolic Engineering, State Key Laboratory of Microbial Resources, Institute of Microbiology, Chinese Academy of Sciences, Beijing 100101, China. [2] University of Chinese Academy of Sciences, Beijing 100049, China. [3] School of Life Sciences, University of Science and Technology of China, Hefei, Anhui 230027, China. [4] Institute of Microbiology and College of Life Sciences, Zhejiang University, Hangzhou, Zhejiang 310058, China. Correspondence and requests for materials should be addressed to Y.Z. (email: zhangyp@im.ac.cn) or to Y.L. (email: yli@im.ac.cn)

A stable and sustainable energy supply is crucial for all life on the earth. As the most abundant and renewable energy source, solar energy can be utilized in the form of electricity through photovoltaics (PV) that are made of semiconductor materials[1,2]. Along with the development of PV materials, the energy conversion efficiency is approaching the theoretical level of 33.7%[3]. However, the toxicity and hard-to-recycle characteristic of PV materials raise concerns on environmental compatibility[3]. Biophotovoltaics (BPV) uses biological photosynthetic materials to convert solar energy into electricity[4]. BPV mainly comprises single species of photosynthetic microorganisms, such as cyanobacteria or eukaryotic microalgae[5,6]. This means BPV is more environmentally compatible and potentially more cost-effective over semiconductor-based PV[7]. It is worth noting that BPV is also a carbon neutral energy production manner as no net carbon dioxide emission occurs during a BPV process. In addition, BPV systems can potentially operate continuously throughout day and night, as the organic compounds accumulated during the day through photosynthesis can be converted to electricity during the night[4,6]. Furthermore, the photocurrent generated in a BPV system can be used to drive the hydrogen fuel production at the cathode[6,8]. Therefore, BPV offers a biological alternative for solar energy utilization.

The BPV process can be divided into two stages, charging and discharging. During the charging stage, the solar photons are first absorbed by the photosystem and the excited electrons generated are stored in high-energy intermediates such as plastoquinol (PQH$_2$), NADPH, or organic compounds resulted from carbon fixation[4,6]. The captured electrons are then exported from the cell interior to the external electrical circuit, thus producing a current, which is the discharging stage[4]. Hence, both photosynthetic capacity and exoelectrogenic activity are required for a BPV system[7]. Conventional BPV systems were mainly developed by immobilizing photosynthetic cells to the anodes for direct extracellular electron transport (EET)[5], or using exogenously added artificial electron mediators for indirect EET[9]. However, the electron exchange processes between cells and electrodes are often hindered by the extremely weak capacity of EET in cyanobacteria or eukaryotic microalgae, and the poor knowledge of endogenous EET mechanism impedes its engineering for improvement[10,11]. Most power outputs of the mediator-less BPV systems described to date have not exceeded a few mW·m$^{-2}$, which is three orders of magnitude lower than the predicted theoretical maximum power output (7 W·m$^{-2}$) of a BPV system working in ambient light[4,12]. Although the addition of artificial mediators can facilitate electron transfer from photosynthetic cells to the anodes, these mediators do not selectively and maximally accept electrons from the photosynthetic electron transfer chain[7,13]. Furthermore, exogenous mediators are potentially toxic to cells and their addition also increases the cost of BPV[7,14–16]. Therefore, the current BPV systems are far from practical applications.

An efficient BPV system needs to be equipped with a highly efficient EET component. The common approach is to introduce an EET pathway into photosynthetic microorganisms, e.g., cyanobacteria, by genetic engineering. To date, no successful attempts have been reported, presumably due to the difficulty in membrane protein expression and translocation, and the poor efficiency in accurately assembling the multiple heme-containing proteins involved in EET pathway[7]. Therefore, it is difficult to incorporate the exoelectrogenic activity into a photosynthetic microorganism. Based on the principle of division of labor, we put forward a general design in which the photosynthetic (or charging) and the exoelectrogenic (or discharging) processes of BPV were assigned to two different microorganisms, with the one responsible for the charging process, and another one responsible for the discharging process. Meanwhile, a highly efficient energy carrier for guiding electron transfer between the two microorganisms is necessary. Such a system may better guide electron delivery from the photosynthetic microorganism to the exoelectrogenic microorganism, thus creating a constrained electron flow from light to the energy carrier, then to electricity.

In this study, a synthetic two-species microbial consortium is designed and constructed for electricity production from light based on the aforementioned concept. The microbial consortium consists of an engineered cyanobacterium and an engineered *Shewanella* (Fig. 1). *Synechococcus elongatus* UTEX 2973, a fast growing cyanobacterium[17], is selected as the charging unit. *Shewanella oneidensis* MR-1, a model metal-reducing microorganism capable of converting organic compounds into electricity using its efficient EET capacity[18], is selected as the discharging unit. D-lactate is selected as the energy carrier in this consortium-style BPV system, because D-lactate is the most favorable carbon source and energy provider of *S. oneidensis* MR-1[19] and can be produced from light and CO$_2$ by engineered cyanobacteria[20]. The constructed D-lactate mediated cyanobacteria-*Shewanella* microbial consortium can produce relatively high-power electricity from light for over 40 days. Our study shows that redirecting the electron flow in a photosynthetic-exoelectrogenic microbial consortium can circumvent the weak exoelectrogenic activity of cyanobacteria, thus improve the efficiency and longevity of BPV.

## Results

**Current production in mono-culture of *S. oneidensis* MR-1.** Dual-chamber electrochemical devices (Supplementary Fig. 14a, b) were used to evaluate the efficiency of current production in mono-cultures and microbial consortia in this study. M9 medium (Supplementary Table 1) supplemented with lactate was used as anodic electrolyte in the mono-culture of strain MR-1. We first experimentally confirmed that there was no cathodic limitation in our setups—increasing the concentration of cathodic electron acceptor, ferricyanide, over 50 mM has little effect on current density, nor does enlarging the size of cathode (Supplementary Fig. 1a, b). The decrease of current density in the later period was mainly due to the gradual consumption of lactate and the decrease of biofilm stability.

To investigate the current production of strain MR-1 on two lactate isomers, 15 mM D-lactate, L-lactate, or DL-lactate were used as electron donors, respectively. The results showed that strain MR-1 can produce current on either isomer, whereby the current density on D-lactate was slightly higher than on L-lactate (Supplementary Fig. 2a). This was consistent with previous reports that strain MR-1 possesses D- and L-lactate dehydrogenases responsible for oxidizing lactate to pyruvate[19], and it preferentially utilizes D-lactate for growth[21]. The maximum current density positively correlated with the concentration of D-lactate and reached saturation at a D-lactate concentration of 5 mM, and an evident current could be detected at a D-lactate concentration as low as 1 mM (Supplementary Fig. 2b, c). We therefore genetically engineered cyanobacteria to synthesize D-lactate by introducing a D-lactate dehydrogenase gene *ldh* in both strains Syn2973 and Syn2973-*omcS*. The resulting strain Syn2973-*omcS*-*ldh*, which carrying the gene *omcS*, produced higher concentration of D-lactate than that of strain Syn2973-*ldh* (Supplementary Fig. 3), but the mechanism remains to be studied. This D-lactate high-producing strain Syn2973-*omcS*-*ldh* was selected for microbial consortium construction.

**Creating an extracellular environment for two species.** The population interactions and functions in a microbial consortium are strongly affected by the extracellular environment[22]. First,

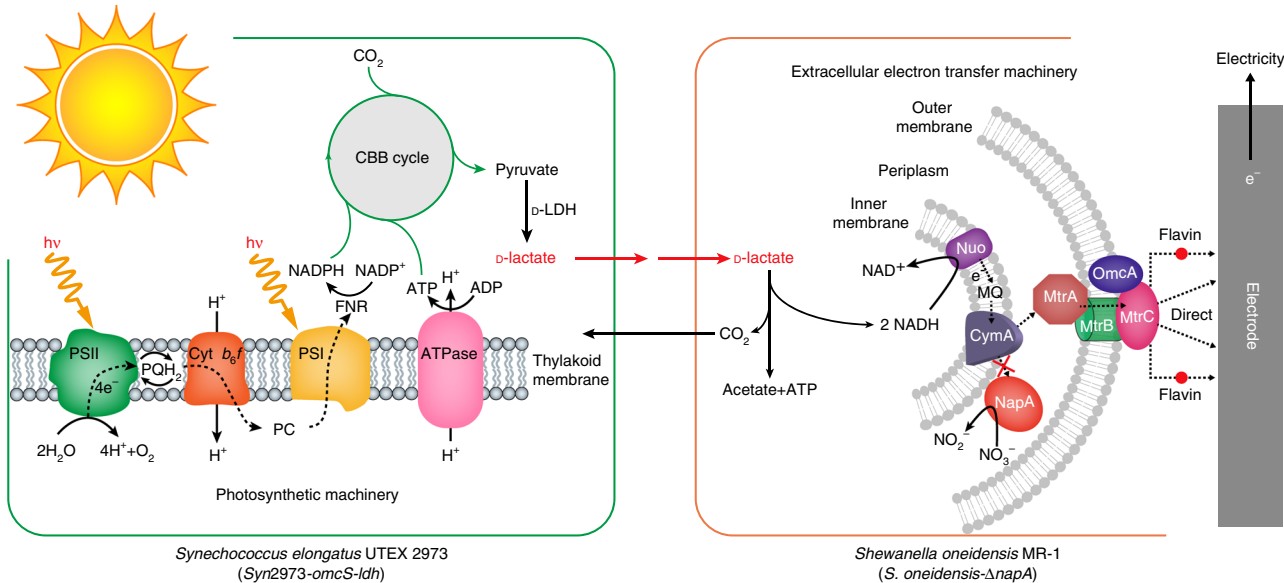

**Fig. 1** Schematic diagram of the BPV system based on a two-species microbial consortium with constrained electron flow. The microbial consortium comprises an engineered cyanobacterium *Synechococcus elongatus* UTEX 2973 (*Syn2973-omcS-ldh*) and an engineered *Shewanella oneidensis* MR-1 (*S. oneidensis-ΔnapA*). Strain *Syn2973-omcS-ldh*, equipped with the photosynthetic machinery and a D-lactate synthesis pathway, harvests light to perform $H_2O$ photolysis and generates high-energy electrons ($e^-$). The electrons are transferred and the subsequently generated NADPH and ATP are used for $CO_2$ fixation and D-lactate production. Strain *S. oneidensis-ΔnapA*, a *napA* deletion mutant, equipped with a D-lactate oxidation pathway and the EET machinery, releases electrons from D-lactate and transfers them to the anode for electricity production. The released $CO_2$ can be re-assimilated by cyanobacteria for D-lactate production. The dotted arrows show the routes of electron transfer in the photosynthetic machinery and EET machinery. PSI/II: photosystem I/II; PQH$_2$: plastoquinol; Cyt $b_6f$: cytochrome $b_6f$ complex; PC: plastocyanin; FNR: ferredoxin-NADPH reductase; ATPase: ATP synthase; CBB cycle: Calvin-Benson-Bassham cycle; D-LDH: D-lactate dehydrogenase; Nuo: NADH:ubiquinone oxidoreductase; MQ: menaquinone; CymA: an inner membrane tetraheme cytochrome c; MtrA: a periplasmic decaheme cytochrome c; MtrB: a non-heme outer membrane porin protein; MtrC/OmcA: two outer membrane decaheme c-type cytochromes; NapA: periplasmic nitrate reductase

developing a suitable medium that satisfies the need of all members of a microbial consortium is necessary[22]. We initially chose the cyanobacterial medium BG11 (Supplementary Table 1) as the candidate. However, strain MR-1 did not produce a current in BG11 (Supplementary Fig. 4a). We identified that buffer salts ($Na_2HPO_4$, $KH_2PO_4$) from M9 are essential to current production for strain MR-1 (Supplementary Fig. 5a). Buffer salts likely act on maintaining the pH and osmotic pressure, however there was still no current produced in BG11 upon addition of buffer salts (Supplementary Fig. 5a). We further identified that two components from BG11, sodium nitrate and ammonium ferric citrate, severely inhibited the current production (Supplementary Fig. 5b). Sodium nitrate and ammonium ferric citrate probably act as the competitive electron acceptors of strain MR-1. Further experiments confirmed that once sodium nitrate and ammonium ferric citrate were removed from BG11, and buffer salts were added simultaneously, a normal current was produced (Supplementary Fig. 5c). For cyanobacteria, the removal of ammonium ferric citrate and the addition of buffer salts did not cause obvious growth defect (Supplementary Fig. 6a). However, sodium nitrate is the sole nitrogen source in BG11 and could not be removed. Therefore, we determined to use a nitrate reductase (NapA) mutant (*S. oneidensis-ΔnapA*), which lost the ability to reduce nitrate[23]. The *napA* deletion did not cause significant growth defect (Supplementary Fig. 6b), but it relieved the nitrate inhibition (Supplementary Fig. 5d). Thus, a modified medium suitable for two species was established, which was named MBG11 (Supplementary Table 1).

We attempted to construct microbial consortium using MBG11. Unfortunately, the consortium did not produce a current under light (Supplementary Fig. 4b). One possibility was that the electrons released from *S. oneidensis-ΔnapA* would be snatched by oxygen evolved during photosynthesis. We indeed found the dissolved oxygen (DO) of the co-cultures under light was much higher than that in the dark (Supplementary Fig. 7a). Moreover, the strain *S. oneidensis-ΔnapA* is sensitive to light and cell death occurred after illumination for 24 h (Supplementary Fig. 7b). Light-induced damage was also reflected in increased intracellular reactive oxygen species (ROS) and oxidation-reduction potential (ORP) (Supplementary Fig. 7c, d). To overcome these problems, we came up with the idea of developing the spatial-temporal separation organization for constructing the cyanobacteria-*Shewanella* consortium (Fig. 2). In temporal separation organization, the charging process (photosynthetic production of D-lactate) and discharging process were implemented sequentially under light and dark conditions, respectively, which enabled discharging to proceed in the oxygen-free and illumination-free environment (Fig. 2a). In spatial separation organization, cyanobacteria and *S. oneidensis* were cultured in two individual chambers, which allowed *S. oneidensis* to be shielded from light independently (Fig. 2b). To sum up, we created a compatible extracellular environment for two species by manipulating it at genetic level, growth medium level and device level.

**Achieving high-power output by temporal separation.** As described above, a defined photosynthetic-exoelectrogenic microbial consortium composed of the engineered cyanobacterium (*Syn2973-omcS-ldh*) and the engineered *S. oneidensis* (*S. oneidensis-ΔnapA*) was constructed in MBG11 through temporal separation organization (Fig. 2a and Supplementary Fig. 14d). At the charging stage, strain *Syn2973-omcS-ldh* produced

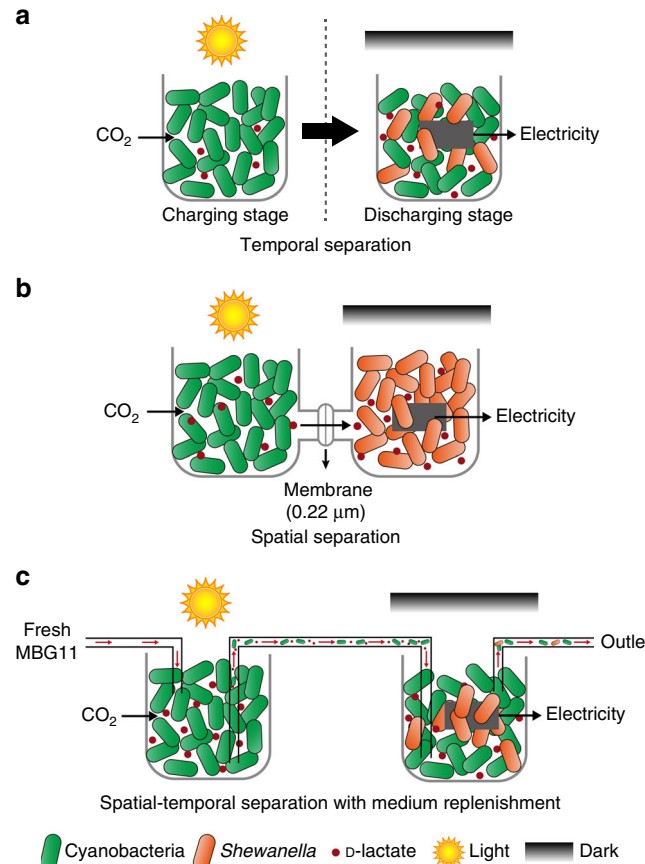

Fig. 2 Schematic illustration of three setups used for constructing microbial consortium. **a** Temporal separation setup. The charging process (photosynthetic production of D-lactate) and discharging process are implemented sequentially under light and dark conditions, respectively. **b** Spatial separation setup. Cyanobacteria and *S. oneidensis* are cultured in two individual chambers separated by a micro-porous membrane, which ensures the charging and discharging processes proceed at the same time and allows *S. oneidensis* to be shielded from light. **c** Spatial-temporal separation setup with medium replenishment. The cyanobacterial culture is replenished with fresh MBG11 medium. This enables a continuous production of D-lactate in the cyanobacterial chamber (left), which is used for electricity production in the anodic chamber (right). Here, all cathodic chambers are omitted

approximately $300 \ \mathrm{mg \cdot L^{-1}}$ of D-lactate in MBG11 for 96 h and it also grew better than the negative control strain *Syn*2973-*omcS* (Supplementary Fig. 8a, b). Then the strain *S. oneidensis-ΔnapA* was inoculated into the anodic chamber together with the culture of strain *Syn*2973-*omcS-ldh* to form a microbial consortium for discharging process, which indeed produced a current of about $300 \ \mathrm{mA \cdot m^{-2}}$ (Fig. 3a). We also confirmed the cyanobacterial strains carrying the gene *omcS* did not produce a significant extra current in the experimental setup used in this study (Supplementary Fig. 9), which ruled out the possibility of direct contribution from the gene *omcS* to current production. Thus, the envisioned cyanobacteria-D-lactate-*Shewanella* (CLS) microbial consortium was developed successfully.

The robustness of microbial consortia is an important consideration for maintaining their function[24]. For a synthetic microbial consortium, the robustness can be evaluated by its resistance to environmental fluctuations and susceptibility to perturbations caused by the variation of inoculated cell density. When the concentration of buffer salts was changed from 1/5 to 1/20 of those in M9, the CLS microbial consortium exhibited

similarly high current densities (Fig. 3b). Furthermore, after gradually reducing the inoculated cell density of strain *S. oneidensis-ΔnapA* from an $OD_{600}$ of 2.0 to 0.01, the CLS microbial consortium maintained functional stability with the average current density remaining at $280 \ \mathrm{mA \cdot m^{-2}}$ for at least 120 h (Fig. 3c). The energy conversion efficiencies (D-lactate to electricity, $\eta_E$) of these BPV systems are pretty high, ranging from 50 to 60% (Fig. 3d). These results illustrated the strong robustness of the CLS microbial consortium against environmental and biological perturbations. In subsequent experiments, we selected a concentration of buffer salts equivalent to 1/5 of that of M9 as the optimal concentration (Supplementary Table 1) and designated an initial $OD_{600}$ of 0.1 as the optimal inoculated cell density of strain *S. oneidensis-ΔnapA*.

In addition to robustness, the CLS microbial consortium efficiently converted light to electricity, with superior performance compared with D-lactate-free or axenic counterparts in terms of productivity and stability (Fig. 4). Under the optimized conditions, a high and stable current was produced by the CLS microbial consortium, which reached up to $350 \ \mathrm{mA \cdot m^{-2}}$ and lasted for 7 days (Fig. 4a). By comparison, the D-lactate-free microbial consortium, consisting of strain *Syn*2973-*omcs* and strain *S. oneidensis-ΔnapA* (named CS microbial consortium), could only produce a low current of about $120 \ \mathrm{mA \cdot m^{-2}}$ (Fig. 4a). Notably, the current density of the CLS microbial consortium was much higher than that of the mono-culture of strain *S. oneidensis-ΔnapA* supplemented with an equal concentration of D-lactate $(300 \ \mathrm{mg \cdot L^{-1}})$, which produced an average current of $150 \ \mathrm{mA \cdot m^{-2}}$ (Fig. 4a). The energy conversion efficiency (D-lactate to electricity, $\eta_E$) of the CLS microbial consortium was 69.5%, which was much higher than that of 26.9% in the mono-culture supplemented with D-lactate. The performance difference between these two systems could be partially ascribed to that additional organics were produced by strain *Syn*2973-*omcS-ldh*, and these organics were used by *S. oneidensis-ΔnapA*. On one hand, we analyzed the HPLC spectrum of the extracellular metabolites of strain *Syn*2973-*omcS-ldh*. Besides the D-lactate which is the dominant secreted metabolite of strain *Syn*2973-*omcS-ldh*, we also found $20–30 \ \mathrm{mg \cdot L^{-1}}$ formic acid was produced (Supplementary Fig. 8c). Formic acid can be used by *S. oneidensis* for current production, this could partially account for the phenomenon we observed. On the other hand, cyanobacteria may release their endogenously stored compounds through dark fermentation and cell lysis during the discharging process[25], which might provide more substrates for *S. oneidensis-ΔnapA* in the CLS microbial consortium. These results suggested the superiority of the CLS microbial consortium in current production.

Electrochemical characterizations were conducted to study the electron transfer efficiency and performance of the CLS microbial consortium and its counterparts. Cyclic voltammetry (CV) at a low scan rate was applied to reveal the kinetics of redox reactions at the cell-electrode interface[26]. A typical redox peak of *S. oneidensis* biofilm starting from around −0.4 V (vs. Ag/AgCl) appeared in the CV curves and the CLS microbial consortium produced the highest catalytic current (Fig. 4b). Polarization curves, obtained from linear sweep voltammetry (LSV), were used to characterize the internal resistance and maximum current density of the corresponding systems. The results showed that the dropping slope of the polarization curve obtained from the CLS microbial consortium was the smallest in comparison with the other systems, indicating less internal resistance (Fig. 4c). Moreover, the maximum current density of the CLS microbial consortium was close to $1400 \ \mathrm{mA \cdot m^{-2}}$ (Fig. 4c). This significant increase on current density at low voltages can be explained from two aspects. On one hand, the electron donor was adequate in the CLS microbial consortium and the cyanobacteria surrounding the biofilm of *S. oneidensis* could produce more D-lactate. Thus,

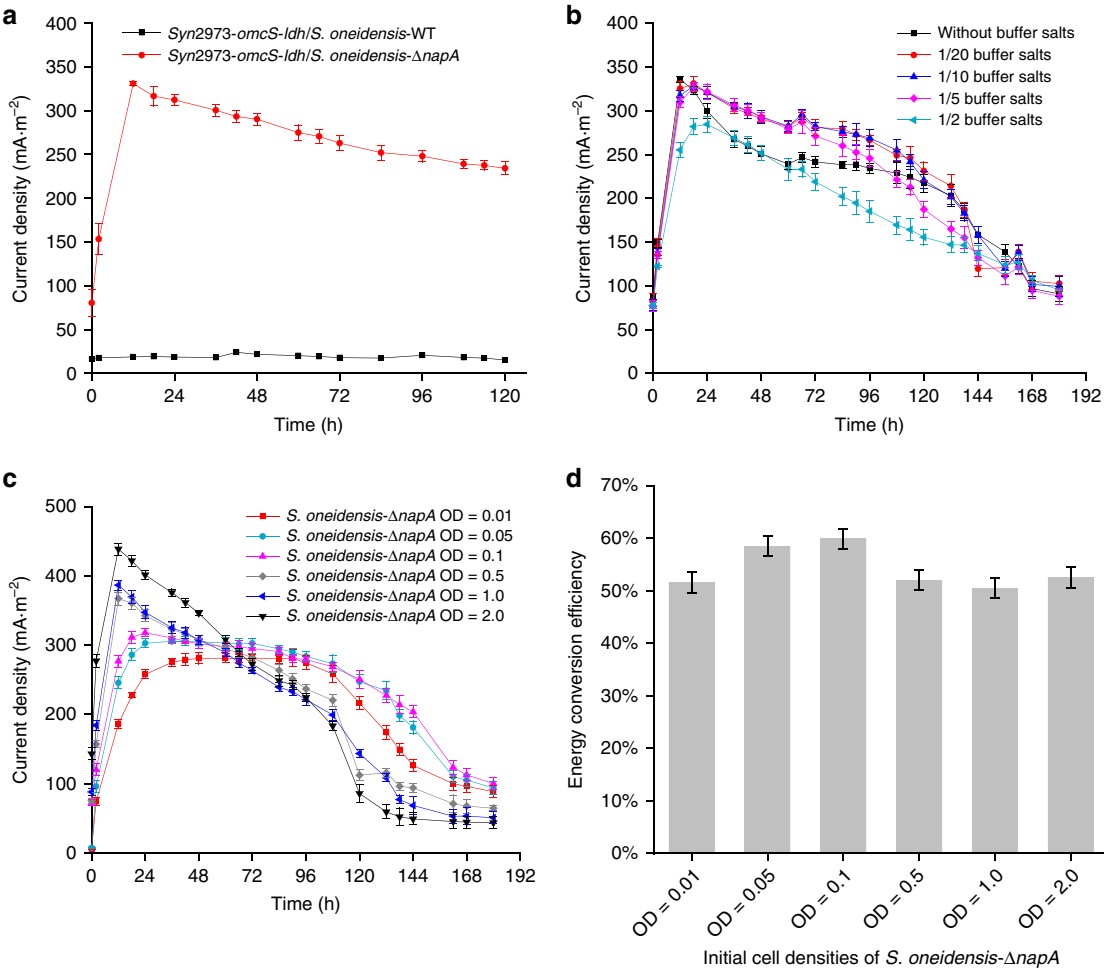

**Fig. 3** Performance of the temporally separated CLS microbial consortium. **a** Current density produced by two microbial consortia comprising *Syn*2973-*omcS-ldh* and *S. oneidensis*-WT or *S. oneidensis*-Δ*napA*, respectively. **b** The robustness of the CLS microbial consortium evaluated in MBG11 medium with different concentrations of buffer salts. **c** The robustness of the CLS microbial consortium evaluated by gradually reducing the inoculated cell density of *S. oneidensis*-Δ*napA*. **d** The energy conversion efficiency (D-lactate to electricity) of the CLS microbial consortium with different inoculated cell densities of *S. oneidensis*-Δ*napA*. All experimental setups were incubated in the dark. Error bars represent the standard deviations from $n = 3$ independent experiments. Source data are provided as a Source Data file

the typical diffusion limitation[27] of electron donor might not exist in the CLS microbial consortium even at low voltages. On the other hand, the oxidation of electron donors would be accelerated and more electrons would be transferred to the anode when the anode potential increased[28,29], that is when the voltage was close to zero V (refer to Methods). Therefore, we speculated the electron transfer process in the CLS microbial consortium would change from the typical diffusional controlled mode to the kinetic controlled mode at low voltages, which would lead to a significant increase on current density. Power curves, derived from the polarization curves, showed the highest power density of 150 mW·m$^{-2}$ for the CLS microbial consortium (Fig. 4d). The maximum power density obtained here is approximately one order of magnitude higher than that of the mediator-less BPV devices with conventional configurations (non-microminiaturized and non-highly advanced anode)[5]. This power output is about one third of the highest BPV power outputs reported to date[30,31], which were achieved through combination of device miniaturization, configuration optimization, optimization of operating conditions, and utilization of highly conductive anode materials.

In view of the large scale of our setup, the CLS microbial consortium was further evaluated by enlarging the size of anode from 2.5 × 2.5 cm to 5.0 × 5.0 cm, gradually. The results showed the current output improved, whereas the current density decreased, along with the enlargement of anode, and the maximum power density is inversely proportional to the anode area in this size range (Supplementary Fig. 10a–c). These results indicated the smaller the size of anode, the higher the current density and power density it produced.

**Achieving longer power output by spatial separation.** To further evaluate the performance of CLS microbial consortium, the microbial consortia were constructed in a three-chamber device under light (Supplementary Fig. 14e). In this spatial separation setup, cyanobacteria and *S. oneidensis* were cultured in two individual chambers, which allowed *S. oneidensis* to be controlled under dark independently (Fig. 2b). A micro-porous membrane between the two chambers enables the diffusion of metabolic products, mainly D-lactate, but prevents two species of microorganisms migrating from one chamber to another (Fig. 2b). At the same time, the oxygen evolved from cyanobacteria during photosynthesis could be isolated to some extent by the microporous membrane, which kept the anodic chamber in a microaerobic environment. This spatially separated CLS microbial consortium was able to maintain a prolonged power output, and thereby sustained current density of about 200 mA·m$^{-2}$ with

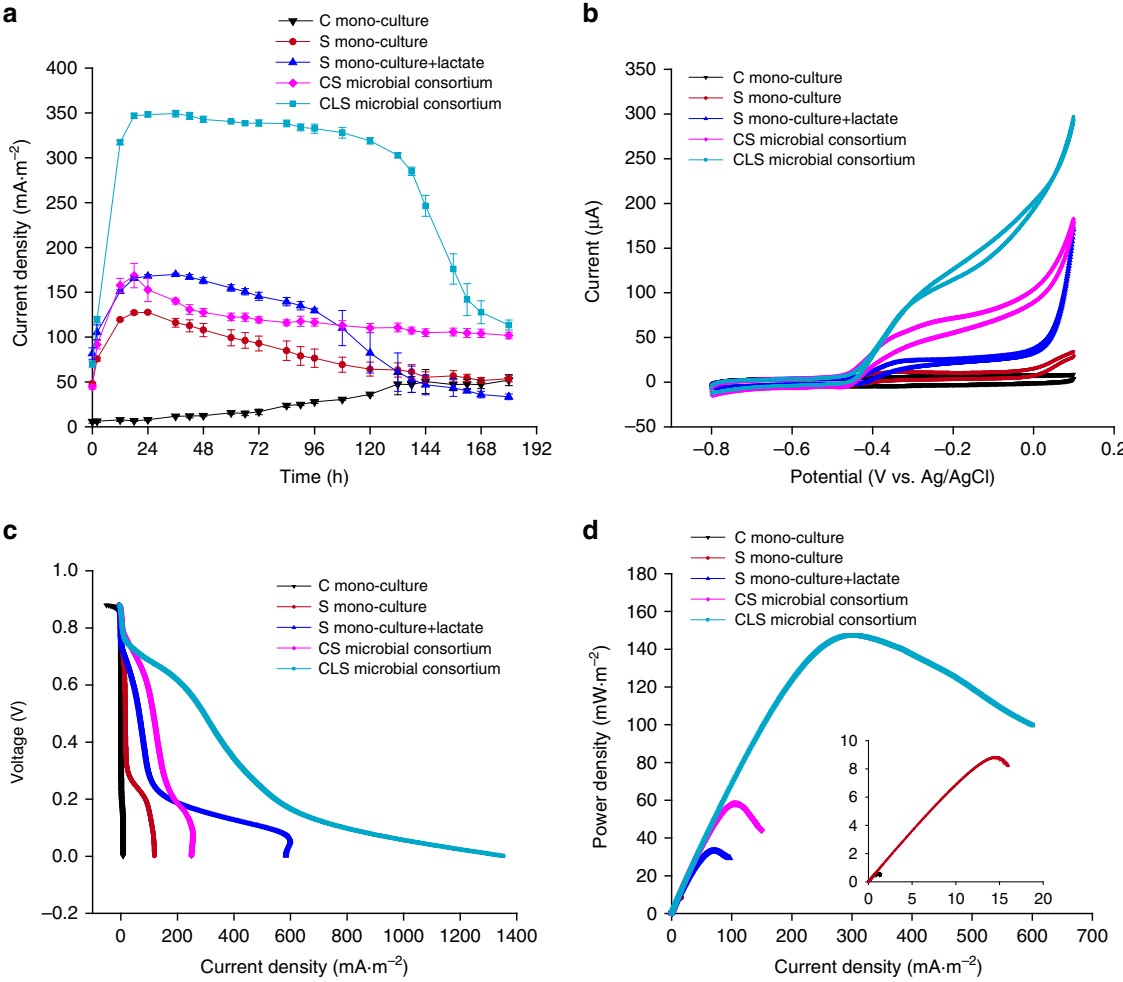

**Fig. 4** Electrochemical characterization of the temporally separated microbial consortia. **a** Current density produced by the CLS microbial consortium and its counterparts. C mono-culture represents the mono-culture of *Syn*2973-*omcS-ldh*, and S mono-culture represents the mono-culture of *S. oneidensis-*Δ*napA*. **b** Cyclic voltammetry (CV) at a low scan rate of 1 mV·s$^{-1}$. **c** Polarization curves obtained from linear sweep voltammetry (LSV) with a low scan rate of 0.1 mV·s$^{-1}$. The voltage in *Y*-axis represents the potential difference between the cathode and the anode. **d** Power curves derived from the polarization curves. Error bars represent the standard deviations from $n = 3$ independent experiments. Source data are provided as a Source Data file

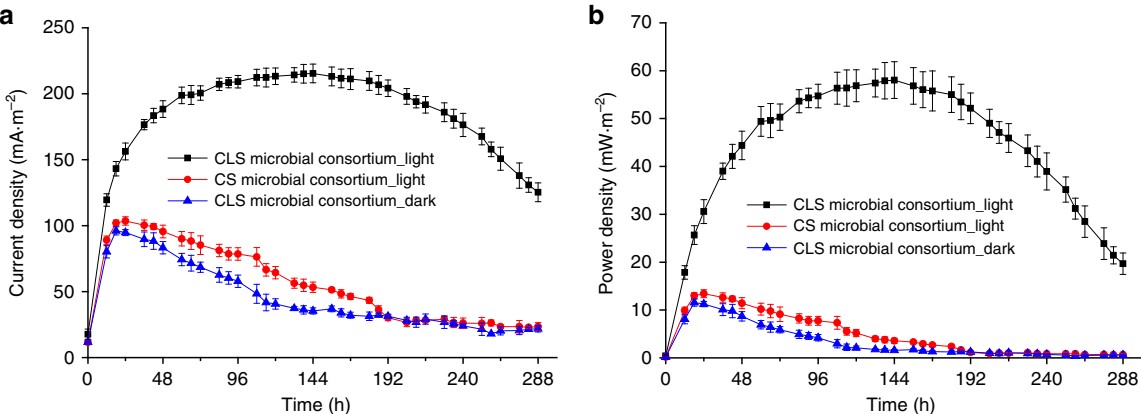

**Fig. 5** Achieving longer power output by the spatially separated CLS microbial consortium. Cyanobacteria and *S. oneidensis* were spatially separated, which allowed D-lactate production under light and current production under dark to be achieved simultaneously. **a** Current density produced by the microbial consortia constructed under light or dark (for cyanobacterial chambers). **b** Power density derived from the current density curves. The anodic chambers containing strain *S. oneidensis-*Δ*napA* were always shielded from light during the whole experimental process for all setups. Error bars represent the standard deviations from $n = 3$ independent experiments. Source data are provided as a Source Data file

power density of around 50 mW·m$^{-2}$ for more than 12 days (Fig. 5a, b). This result indicated that the confined spatial structure stabilized the microbial community and provided a better environmental control for both microorganisms.

The CLS microbial consortium worked efficiently via syntrophic metabolism, which was not only reflected in the high efficiency of power output, but also in robust cell growth. The heterotrophic microorganism, strain *S. oneidensis-ΔnapA*, enhanced the growth of cyanobacteria by 6–35% over the axenic cultures (Supplementary Fig. 11). It is worth mentioning that the growth enhancement in the CLS microbial consortium was much more pronounced than in the CS consortium (Supplementary Fig. 11). These results further emphasized the significance of a specific energy carrier which rerouted the electron flow for a synthetic microbial community.

**Long-term power output by spatial-temporal separation**. The application of a BPV system not only depends on how high the power output is, but also depends on how long and how stable the system operates. Although the period of current production was prolonged by stabilizing microbial consortium through spatial separation, the power output still decreased to a low level after 12 days (Fig. 5b). Perhaps this is because the cyanobacterial cells were no longer producing D-lactate, which could be ascribed to the deficiency of minimal nutrients.

To test this hypothesis, we developed a spatially temporally separated CLS microbial consortium through replenishing the minimal nutrient for cyanobacteria in the dual-chamber device (Fig. 2c and Supplementary Fig. 14f). In this setup, the fresh MBG11 medium was replenished to an accessional bottle used for the cultivation of cyanobacteria, at a constant flow rate of about 50 mL·d$^{-1}$. The cyanobacterial culture was then delivered into the anodic chamber which containing *S. oneidensis-ΔnapA*, at the same flow rate, for building the microbial consortium and producing current. Meanwhile, the anodic electrolyte was also flowed out from the anodic chamber at the same flow rate. The charging process occurred in the cyanobacterial bottle and the discharging process occurred in the anodic chamber, which constitute a spatial-temporal separation setup. The cell density of cyanobacteria at OD$_{730}$ was controlled between 3.0 and 4.0, which was a suitable density in favor of D-lactate production according to the preceding result (Supplementary Fig. 8). By monitoring the D-lactate production throughout the course, the concentration of D-lactate maintained at 2.5–4.0 mM (Fig. 6), which was sufficient to support the current production of *S. oneidensis* (Supplementary Fig. 2c). Strikingly, this flow setup could stably operate for more than 40 days with a high current density of 240–380 mA·m$^{-2}$, achieving a substantially prolonged current production (Fig. 6).

Initially, during the period of 0–13 days, 5% LB medium was added to the anodic chamber with the aim to regenerate *S. oneidensis* cells and maintain its activity, but the current density was neither very high nor stable. Moreover, we observed excess growth of *S. oneidensis* in the anodic chamber and the inside of adjacent silicone tubes, which might result in consumption of much D-lactate for its growth and respiration, thus decreased the proportion of D-lactate used for current production. Therefore, we empirically decreased the addition of LB to 1% from the 13th day onward. Interestingly, the current production was gradually restored and the current density was maintained at a high level of ~380 mA·m$^{-2}$ for over 20 days, stably (Fig. 6). It is likely that when LB addition decreased from 5 to 1%, the steady state of biofilm of *S. oneidensis* was disturbed and the biofilm needed to be reconstructed so as to reach a newly established steady state. This process lasted for seven days, and the profile is similar to

that of the process at the initial stage (Fig. 6, 0–3rd days), during which the current density increased continuously along with the gradual formation of the biofilm. A further experiment confirmed that 1% LB addition, rather than 5 or 0%, could maintain the cell density of *S. oneidensis* at a stable level, thereby stabilizing the current production at a higher level (Supplementary Fig. 12a, b). The results showed when LB addition was 5%, the current density first increased then started to continuously decline, whereas when LB addition was 1%, the current density continuously increased (Supplementary Fig. 12a). This feature can help us to better understand the continuous increase of current density during the 13–20th days in Fig. 6 when the LB addition was decreased from 5 to 1%. Conceivably, the higher and stable current density could also be partially ascribed to the higher D-lactate production from the 12th day onward (Fig. 6). The calculated average current density and average power density of the whole process reached a relatively high level of about 320 mA·m$^{-2}$ and 135 mW·m$^{-2}$, respectively (Fig. 6). This demonstrated that when the *S. oneidensis* cells were retained stably in the anodic chamber, a sustained production of D-lactate by cyanobacteria upon medium replenishment could lead to an efficient and longevous power output in this CLS microbial consortium-based BPV system.

## Discussion

Since BPV utilizes the biological function of living organisms, we argue that biotic engineering approaches may help better address the bottleneck of BPV systems—the low EET efficiency of photosynthetic microorganisms. In this study, we created a BPV system which used an energy carrier mediated microbial consortium comprised of an exoelectrogenic microorganism to circumvent the low EET efficiency of cyanobacteria. The selection of a specific energy carrier for this consortium, D-lactate, effectively enhanced the electron flow from photosynthetic microorganism to exoelectrogenic microorganism, and ultimately the electrons were collected by electrode for current production. By mimicking nature, the temporal/spatial separation design conquered the physiological incompatibility between the members of the microbial consortium. In the temporal separation setup, the CLS microbial consortium produced a power density of over 150 mW·m$^{-2}$ and maintained it for 7 days (Fig. 4), this power density is approximately one order of magnitude higher than mediator-less BPV devices with conventional configurations. In the spatial separation setup, the CLS microbial consortium sustained a longer power output under illumination for more than 12 days (Fig. 5). Most interestingly, in the spatial-temporal separation setup with minimal medium replenishment, the CLS microbial consortium achieved a steady power output, 135 mW·m$^{-2}$ on average, for over 40 days (Fig. 6). Comparison with existing BPV systems are summarized in Supplementary Data 1.

Thanks to the improvement on bacterial adhesion, electrode conductivity, mass transport, and decrease of internal resistance, the maximum power density of a BPV system has been improved by 500–1000-fold in the last decade (Supplementary Data 1). In the previous studies, abiotic engineering approaches were used to enhance electron transfer in biotic-abiotic interface, which mainly addressed how to efficiently transfer electrons from cell surface to the electrodes. This includes electrode engineering[31–36], device miniaturization/optimization[30,31,37], and mediator diversification[9,13,15,16,38–40]. To date, the highest power density reported was 500 mW·m$^{-2}$ for BPV systems, but the longevity of actual continuous operation was not described[30]. This power density was achieved by decoupling the charging and power delivery process, which allowed these two core processes to be optimized independently[30]. Ultimately, the microscale design

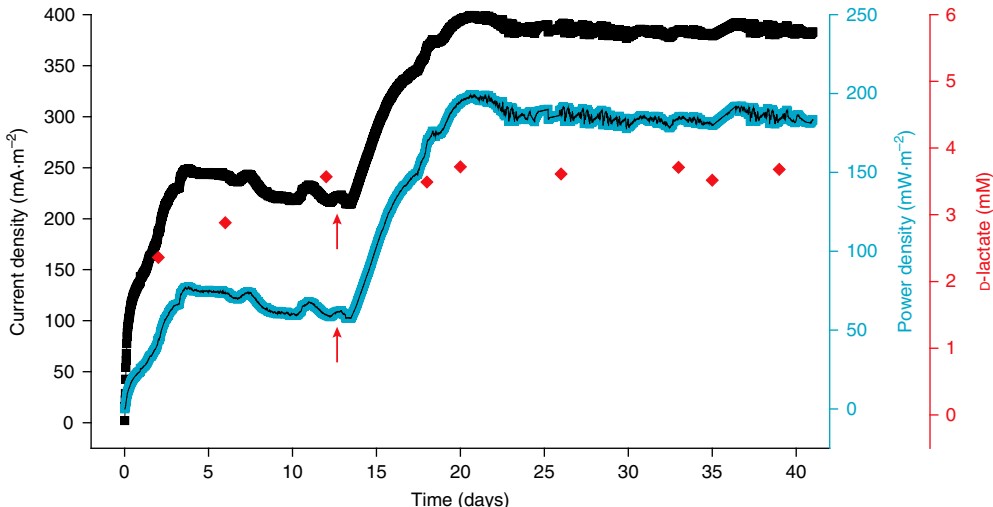

**Fig. 6** Achieving long-term steady power output by the spatially temporally separated CLS microbial consortium with medium replenishment. Electrical outputs (current density and power density) from the CLS microbial consortium were shown. The concentration of D-lactate in the cyanobacterial culture was measured at several time points. During the period of 0–13th days, 5% LB medium was added to the anodic chamber. From the 13th day onward, the addition of LB medium was decreased to 1% (red arrows indicated). Current density was recorded every 20 min. The data shown in the different curves were from one of the experiments, each contains more than 2800 data points. Source data are provided as a Source Data file

together with the flow-based operation, the use of electron mediator and cyanobacterial mutant increased the power density to a high level[30]. The maximum power density of our system is one-third of that of the miniaturized BPV device[30]. Nevertheless, the total power output of our device (93.75 µW) is much larger than that of the miniaturized BPV device (0.04 µW) (Supplementary Data 1). This means ~2300 miniaturized BPV devices are required to work simultaneously to reach an equal power output achieved by our system. We also confirmed that a common electron mediator, potassium ferricyanide, is sensitive to white light and would be decomposed to $Fe(OH)_3$ and KCN, once exposed to full spectrum fluorescent lamps (Supplementary Fig. 13a)[41]. Under the same illumination condition, the cyanobacterial cells died after 3 days when grown with ferricyanide even at a low concentration of 1 mM, which probably due to the toxicity of the decomposed product KCN of ferricyanide (Supplementary Fig. 13b). Previous BPV studies did not specifically describe this phenomenon, which is probably due to the short operating time under illumination conditions, or the use of monochromatic light which is less energetic compared with the one used in our study. Some other reports also mentioned the exogenous electron mediators used in BPV are harmful for microorganisms[7,15,16,36].

The lifespan is equally important to the power density of a BPV system. However, the long-term operation was omitted in most BPV systems (Supplementary Data 1). To date, only a few systems could maintain more than 10 days at an extremely low power density (<0.2 mW·m$^{-2}$). A miniaturized biological solar cell was developed via creating a 3-D conductive anode and configuring with gas-permeable microfluidic system. This allowed the current production sustain for about 20 days with a light/dark cycle (average power density was about 120 mW·m$^{-2}$)[31]. Our system could maintain a more steady current production for over 40 days at a power density of 135 mW·m$^{-2}$, on average. This high longevity was achieved through the continuous supply of inorganic nutrients, which ensured the continuous production of energy carrier, D-lactate, from photons. Two types of BPV were developed previously[4,30]. They either relied on suspending photosynthetic cells in solution supplemented with artificial mediators, or relied on immobilizing cells to the anodes. Neither of these conditions is beneficial for the growth and metabolism of photosynthetic microorganisms, which might prevent its long-

term operation. In our system, the photosynthetic microorganism was cultured in a suitable medium, in a planktonic form, without addition of potentially toxic artificial mediator. The photosynthetic microorganism may well in an active status, which might be one of the key reasons contributed to the observed longevity. In addition, different from the previously observed long-term current production achieved in a microscale device (90 µL) which relied on biofilm formation[31], our system is based on a large suspension (140 mL) of microorganisms. The fact that such a steady and longevous current production was achieved in a 100-mL scale system demonstrates BPV has the potential to stably supply electricity in a larger scale and a wider space.

The interactions between members of microbial consortia rely on chemical communication[42]. Therefore, engineering chemical communication becomes particularly important for designing a synthetic microbial consortium. In this study, D-lactate was rationally chosen as an efficient energy carrier, which is responsible for chemical communication and unidirectional energy transfer between cyanobacteria and *Shewanella*. The engineered strain *Syn2973-omcS-ldh*, which was selected as the light-harvesting microorganism, produced a relatively high yield of D-lactate. According to the calculation of energy allocation, 10.3% of total fixed light energy was shunted into D-lactate used for current production (Fig. 7). D-lactate can easily be metabolized to form NADH by *S. oneidensis*, and the released electrons can be transferred outward into the anode with a high efficiency of about 70% (Fig. 7). Hence, a constrained electron flow (photons → NADPH/ATP → D-lactate → NADH → electricity) was established in our system. This constrained electron flow dramatically improved the performance of two-species BPV relative to a low power output in the D-lactate-free system (Fig. 4). Therefore, the defined microbial species and constrained electron flow jointly contribute to the efficient and longevous power output of this photosynthetic-exoelectrogenic microbial consortium. Compared with single species of cyanobacteria with low EET efficiency, more electrons generated through water photolysis were ultimately exported from photosynthetic cells to the anode in the CLS microbial consortium (Figs. 4a and 7). Based on relatively clear electron flow towards anode, our system holds the potential for further improvement on power density and longevity. From the perspective of microbial community engineering, developing a

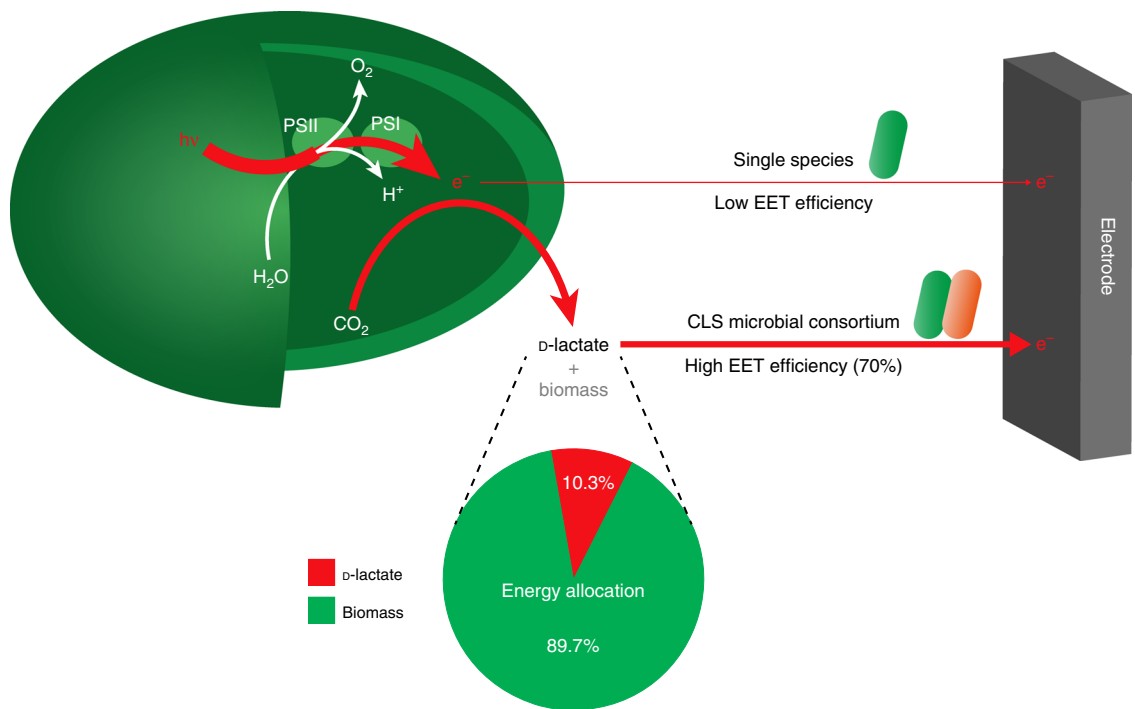

**Fig. 7** Electron flow comparison of two different types of BPV. One type is based on the single species of cyanobacteria, and another is based on the CLS microbial consortium. The red solid arrows show the routes of electron transfer and the thickness of the arrows indicates the relative flux of electrons. In the single species of cyanobacteria, photoexcited electrons are transferred directly to the electrode with a low EET efficiency. In the CLS microbial consortium, photoexcited electrons are diverted to D-lactate by the engineered cyanobacterium, and *S. oneidensis* releases these electrons to the electrode with a high EET efficiency of about 70%. Solar energy allocation in the strain *Syn2973-omcS-ldh* was calculated based on the heat of combustion and shown at the bottom. Source data are provided as a Source Data file

programmed regulatory circuit for microorganisms to control their behaviors, for instance, charging in the day and discharging only in the night, provide opportunities for tighter control between the members of microbial consortium for intelligent power output[43,44].

The syntrophic interaction between cyanobacteria and *Shewanella* probably exists in the natural environment. As ecologically important primary producers, marine cyanobacteria account for a substantial fraction of organic carbon in the oceans[45,46]. The known species of the genus *Shewanella* are widely distributed in marine and fresh-water environments[47]. Most of the species were isolated from deep-sea sediments, and *S. oneidensis* MR-1 was isolated from the sediment of Oneida Lake[47–49]. The deep-sea sediments are rich in organic and inorganic electron acceptors that can be used for respiration by *Shewanella* oxidizing organic carbon for growth. The water movement of the oceans drives nutrient exchange processes among different species. Cyanobacteria and *Shewanella* are separately distributed from the surface to the depth of marine regions, which forms a simple social network of microorganisms across space and time, contributing to the biogeochemical cycles. From this perspective, the cyanobacteria-*Shewanella* consortium constructed here is a remodeling of natural microbial ecosystems. These indicated that there is enormous potential for the utilization of natural autotrophic-heterotrophic microbial communities in energy and environmental applications.

In conclusion, we designed and created a robust, efficient, and durable BPV system driven by an energy carrier mediated microbial consortium, guided by the constrained electron flow concept. This study significantly advances our understandings on the efficiency and longevity of BPV, thus represents an important step toward further improvement of BPV systems. Furthermore, this microbial consortium-driven BPV prototype can also serve as

a platform to study the biological conversion of solar energy into electricity.

## Methods

**Strains and culture conditions**. The wild-type of *S. oneidensis* MR-1 (ATCC 700550) and its *napA* deletion mutant (*S. oneidensis*-Δ*napA*) were kind gifts from the Yong's lab and Gao's lab, respectively[23,50]. *S. oneidensis* strains were cultured in Luria-Bertani (LB) medium at 30 °C. For preparation of mono-cultures and microbial consortia, *S. oneidensis* was cultured in LB medium at 200 rpm for 12 h. The cyanobacteria strains, including *Syn2973*, *Syn2973-omcS*, *Syn2973-ldh*, and *Syn2973-omcS-ldh*, were cultured in BG11 or MBG11 media at 38 °C at 140 rpm in a light incubator with 3% $CO_2$ and a continuous illumination of 150 μmol·m$^{-2}$·s$^{-1}$, unless stated otherwise. All other experiments need to be performed under light were also conducted in the light incubators with the same conditions. The light sources of the incubators are full spectrum fluorescent lamps.

**Construction of cyanobacterial mutants**. The mutants of cyanobacteria, *Syn2973-omcS*, *Syn2973-ldh*, and *Syn2973-omcS-ldh*, were constructed by integrating *omcS* gene or *ldh* gene into the chromosome of *Syn2973* through homologous recombination[17]. Specifically, the plasmid pSyn2973-omcS was used to integrate the *omcS* gene into the *glcD1* site in the chromosome of *Syn2973*, and the plasmid pSyn2973-ldh was used to integrate the *ldhA* gene into the *nblA* site. The plasmid pSyn2973-omcS was constructed by inserting the Prbcl-omcS-Trbcs-Kan expression cassette, which consisted of the promoter Prbcl, the coding region of *omcS* from *Geobacter sulfurreducens*, the terminator TrbcS, and a kanamycin resistance gene (Kan), into the plasmid pBR322 flanked by the sequences homologous to the up- and downstream fragments of gene *glcD1*. The plasmid pSyn2973-ldh was constructed by inserting the Pcpc560-ldh-TrbcS-Cm expression cassette, which consisted of a strong promoter Pcpc560[51], the coding region of gene *ldh* of D-lactate dehydrogenase from *Lactobacillus delbrueckii* ATCC 11842, the terminator TrbcS, and a chloramphenicol resistance gene (Cm), into the plasmid pBR322 flanked by the sequences homologous to the up- and downstream fragments of gene *nblA*. The plasmids were constructed in *Escherichia coli* DH5α. The transformation of plasmids was conducted through tri-parental conjugation[17]. The gene integrations occurred through chromosomal homologous recombination[17].

**Electrochemical devices**. Dual-chamber electrochemical devices (140 ml working volume, Supplementary Fig. 14a–d) separated by Nafion 117 proton exchange

membranes (DuPont, USA) were used for constructing mono-cultures and temporally separated microbial consortia. Carbon cloth (CeTech, Taiwan, China) was used as the anode (2.5 cm × 2.5 cm) and the cathode (3.0 cm × 3.0 cm). The cathodic electrolyte was composed of 50 mM $K_3[Fe(CN)_6]$ in 50 mM potassium phosphate buffer, pH = 7.0. To measure the current production, the anode and the cathode were connected by a 2.0 kΩ external resistor ($R$) (Supplementary Fig. 14c). The voltage ($U$) across the external resistor was measured using a VC980 + digital multimeter (VICTOR, China) or 2638 A Data Acquisition System (FLUKE, WA, USA). The current density ($I$) was calculated according to Ohm's law of $I = U/R$ and normalized to the geometric area of the anode (6.25 cm$^2$). The power density was calculated as $P = U \times I$.

Three-chamber devices (Supplementary Fig. 14e) were used for constructing spatially separated microbial consortia. The anodic chamber (160 mL working volume) was located in the middle. The cathodic chamber (140 mL working volume) and the cyanobacterial chamber (140 mL working volume) were flanking it at the two sides. The cyanobacterial chamber and the anodic chamber were separated by a micro-porous membrane with 0.22-μm pores. The other configurations were the same as the dual-chamber device.

**Constructing mono-cultures for current production.** The mono-cultures of *S. oneidensis* for current production in dual-chamber devices (Supplementary Fig. 14c). After cultivation at 30 °C for 12 h, the *S. oneidensis* cells were harvested by centrifugation at 6000 × *g* for 5 min and re-suspended in the designated medium plus 5% LB as anodic electrolyte, with an OD$_{600}$ of 0.8 unless specified otherwise. The anodic electrolyte was supplemented with 15 mM sodium lactate as electron donor unless otherwise stated. The devices were incubated in the dark incubators at 30 °C without shaking, and the voltages generated were measured using the multimeter.

**Constructing microbial consortia for current production.** The temporally separated microbial consortia were constructed in dual-chamber devices (Supplementary Fig. 14d). The cyanobacteria strain was first cultured in BG11 medium for 48 h, and then the cells were harvested by centrifugation and re-suspended in MBG11 medium to an initial OD$_{730}$ of 0.5 for D-lactate production for 96 h. Then 135 mL cyanobacterial culture plus 5% LB was transferred into the anodic chamber, and *S. oneidensis* cells were subsequently inoculated at an OD$_{600}$ of 0.1 unless specified otherwise. The devices were incubated in the dark incubators at 30 °C without shaking, and the voltages generated were measured using the multimeter.

The spatially separated microbial consortia were constructed in three-chamber devices (Supplementary Fig. 14e). The cyanobacteria strains were cultured in the cyanobacterial chamber (140 mL MBG11) at an initial OD$_{730}$ of 1.0. The strain *S. oneidensis-ΔnapA* was inoculated into the anodic chamber (160 mL MBG11 plus 5% LB) at an initial OD$_{600}$ of 0.1 at the same time. The anodic chamber and cathodic chamber were covered with the tin foil to shield them from light. The cyanobacterial culture was stirred at 250 rpm. The devices were incubated in the light incubators with 3% $CO_2$ at 30 °C, and the voltages generated were measured using the multimeter.

The spatially temporally separated microbial consortia with medium replenishment were constructed in dual-chamber devices plus a 500-mL bottle (Supplementary Fig. 14f). The strain *Syn2973-omcS-ldh* was cultured in the bottle (200 mL working volume) with a stirring at 200 rpm. The fresh MBG11 was continually pumped into the bottle through a silicone tube with a flow rate of 50 mL·d$^{-1}$. Meanwhile, the cyanobacterial culture was pumped out from the bottle and injected into the anodic chamber with the same flow rate. The anodic electrolyte was also pumped out from the anodic chamber with the same flow rate. The strain *S. oneidensis-ΔnapA* was inoculated into the anodic chamber at an initial OD$_{600}$ of 0.2. The anodic and cathodic chambers were shielded from light. The cathodic electrolyte was composed of 200 mM $K_3[Fe(CN)_6]$, which ensures the adequate supply of electron acceptor. The devices were incubated in the light incubators with 3% $CO_2$ at 30 °C, and the voltages generated were recorded using the Data Acquisition System every 20 min.

**Electrochemical analysis.** For the CLS microbial consortium and its counterparts, polarization curves were obtained by performing linear sweep voltammetry (LSV) analysis. LSV analysis was conducted on a two-electrode configuration[52], in which the working electrode was connected to the anode, whereas both the counter electrode and the reference electrode were connected to the cathode. LSV analysis in a two-electrode configuration represents the overall performance of a bioelectrochemical cell[52]. The applied potential on working electrode was changed from the open circuit potential (OCP, −880 mV) to 0 mV at a rate of 0.1 mV·s$^{-1}$ controlled by a CHI1030C potentiostat (CH Instruments, China), and the current ($I$) was recorded in real time. The absolute value of voltage ($U$) between the anode and the cathode was set as the Y-axis of polarization curves. The output power ($P$) was derived via the relationship: $P = U \times I$. The current and power were normalized to the geometric area of the anode (6.25 cm$^2$) to obtain the current density and power density, respectively. Cyclic voltammetry with a low scan rate (1 mV·s$^{-1}$) was conducted in a three-electrode configuration in the potential range from −800 to + 200 mV vs. an Ag/AgCl reference electrode.

**Calculation of energy conversion efficiency.** The energy conversion efficiency of D-lactate to electricity ($\eta_F$, also called Faraday efficiency) was calculated as the ratio of the actual amount of Coulombs generated ($Q$) to the maximum amount of Coulombs released by the oxidation of D-lactate to acetate ($Q_{max}$), using Eq. (1):

$$\eta_F = \frac{Q}{Q_{max}} = \frac{\int_0^T i\, dt}{n\, m\, F} = \frac{\int_0^T U\, dt}{n\, m\, F\, R} \tag{1}$$

where $U$ (V) is the measured voltage of the systems, $R$ is the external resistor (2000 Ω), $n$ is the number of electrons released when one molecule of D-lactate is oxidized ($n = 4$), $m$ (mol) is the molar amount of D-lactate consumed, and $F$ is the Faraday constant (96,485 C·mol$^{-1}$ of electrons). The value of $\int_0^T U\, dt$ is the integral area of the $U$–$t$ curve and $T$ (s) is the period of measurement.

**Solar energy allocation in *Syn2973-omcS-ldh*.** Cyanobacteria directly convert inorganic carbon plus the solar energy into secreted products and biomass. In the strain *Syn2973-omcS-ldh*, D-lactate is the dominant product. Of the others, only a tiny amount of formic acid was detected in the HPLC spectrum. We thus assumed that the fixed inorganic carbon and fixed solar energy by cyanobacteria were totally stored in D-lactate and biomass.

Solar energy allocation into D-lactate ($E_p$) was calculated according to the heat of combustion (also known as the heating value or the calorific value). The standard heat of combustion ($\Delta_c H°$) of D-lactate is 15.13 kJ·g$^{-1}$ from the book of *The Merck Index: An Encyclopedia of Chemicals, Drugs, and Biologicals*, 15th edition[53]. The heating value of biomass of cyanobacteria (*S. elongatus*) is 19.25 kJ·g$^{-1}$ from a recent study[54]. The OD$_{730}$ to dry cell weight (DCW) conversion was determined to be 0.34 gDCW·L$^{-1}$ per OD$_{730}$ for *S. elongatus* UTEX 2973[55]. Thus, the solar energy allocation into D-lactate was calculated using Eq. (2):

$$E_p = \frac{15.13 \times M}{15.13 \times M + 19.25 \times 0.34 \times OD_{730}} \times 100\% \tag{2}$$

where $M$ is the titer of D-lactate (g/L). The value of solar energy allocation into biomass equals to $1 - E_p$.

**Quantification of lactate.** The concentration of lactate in the cyanobacterial cultures was analyzed using an Agilent 1260 HPLC system (Agilent Technologies, CA, USA), equipped with a Bio-Rad HPX-87H column (Bio-Rad Laboratories, CA, USA) kept at 55 °C, with 5 mM $H_2SO_4$ as the mobile phase at a flow rate of 0.5 mL·min$^{-1}$. The injection volume is 10 μL.

**Measurement of DO level and ORP level.** Mono-cultures (*S. oneidensis* only) were constructed for ORP measurement and co-cultures were constructed for DO and ORP measurements. After cultivation in either light or dark incubators, the DO level and ORP level were measured separately using a handheld Microprocessor Pen Type Dissolved Oxygen Meter (Aladdin, Shanghai, China) and a handheld Waterproof Pen Type ORP Meter (Aladdin, Shanghai, China) by immersing the probes into the cell cultures and the results were displayed directly.

**ROS detection and plate drop test.** For evaluation of light-induced damage to the *S. oneidensis* cells, intracellular ROS levels were determined using the Fluorometric Intracellular ROS Kit (Genview, Florida, USA). Briefly, the *S. oneidensis* cells were suspend in MBG11 medium to an OD$_{600}$ of 0.5 and cultured in either light or dark incubators at 30 °C at 130 rpm. At the indicated time points, 100 μL samples were mixed with 100 μL DCFH-DA (ROS Detection Reagent, 20 μM), and the mixtures were incubated at 30 °C for 2 h and the fluorescence intensity was measured at excitation and emission wavelengths of 490 and 520 nm, respectively. The ROS level is positively correlated with the fluorescence intensity[56,57].

The plate drop test was employed to evaluate the viability of *S. oneidensis* cells. Briefly, the *S. oneidensis* cells were cultured in MBG11 medium under either light or dark incubators at 30 °C at 140 rpm with an initial OD$_{600}$ of 0.25, which was set as the undiluted culture (dilution factor 0), and 10-fold serial dilutions were prepared with sterile water. Five microliters of each dilution was dropped onto LB plates. The plates were incubated in the dark at 30 °C for 24 h for colony formation.

**Potassium ferricyanide decomposition under illumination.** To investigate the reaction of potassium ferricyanide under illuminaton, the potassium ferricyanide solutions of 2 mM and 50 mM in flasks were incubated in the light incubator at 30 °C for 12 h without shaking. For control experiments, the flasks were covered with the tin foil. Besides, the strain *Syn2973-omcS-ldh* was cultured in BG11 medium at 30 °C at 140 rpm under the same illumination condition with the addition of 0, 1, 5, 10, and 50 mM potassium ferricyanide, and cell density at OD$_{730}$ was measured.

**Reporting summary.** Further information on research design is available in the Nature Research Reporting Summary linked to this article.

## Data availability

Data supporting the findings of this work are included within the paper and its Supplementary Information files. A reporting summary for this Article is available as a Supplementary Information file. The datasets generated and analyzed during the current study are available from the corresponding authors upon request. The source data underlying Figs. 3–7, and Supplementary Figs. 1–12 and 13b are provided as a Source Data file.

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

## Acknowledgements

This work was supported by the Key Research Program of the Chinese Academy of Sciences (ZDRW-ZS-2016-3) and the National Natural Science Foundation of China (31870038, 31670048, and 31470231). We thank Prof. Yangchun Yong from Jiangsu University for providing the strain *S. oneidensis* MR-1.

## Author contributions

H.Z., Y.Z., and Y.L. conceived the project. H.M., W.Z., J.Z., and H.G. constructed the strains. H.Z. performed all other experiments and analyzed the data. H.Z., Y.Z., and Y.L. wrote the paper. All authors read and approved the final version of paper.

## Additional information

**Competing interests:** The authors declare no competing interests.

