## [Peer Review File · Nature Communications]

Reviewers' comments:

Reviewer #1 (Remarks to the Author):

In this work by Li et al., a “dual cell” biophotovoltaic setup is described. In the first cell, a cyanobacterial culture is grown under illumination. This cyanobacterial strain is genetically modified to produce lactate, which is then transferred to the second ‘dark’ cell, where a strain of *Shewanella oneidensis* MR-1 is grown and the carbon source is converted to electricity.

I cannot recommend this manuscript for publication in Nature Communications due to a wide variety of reasons:

1) Although the authors have nicely described the experimental and scientific reasons why a “dual cell” setup is necessary (rather than a co-culture), the scientific novelty of the “dual cell” setup is minor. Microbial fuel cells using *S. oneidensis* MR-1 mono-cultures with lactate as a carbon source have been extensively described in literature. Similarly, culturing of cyanobacteria and, in particular, *Synechococcus elongatus* have also been extensively described, as is the genetic modification of *S. elongates* for bioenergy applications. The idea of genetically modifying *S. elongatus* to synthesize lactate has also been described by several groups (e.g., [10.1038/srep09777](https://doi.org/10.1038/srep09777), [10.1016/j.jbiosc.2017.02.016](https://doi.org/10.1016/j.jbiosc.2017.02.016)). Thus, the only novelty in this work is combining both established platforms into a combined setup. I note that, after extensive improvement to the manuscript, the work might be of interest as it could set a new benchmark for biophotovoltaics. However, such publication might be better suited for a more specialised journal or potentially for Scientific Reports.

Response:

Thanks for your comment. The scientific novelty of this work is not the simple combining of *Synechococcus elongatus* and *Shewanella oneidensis*. Rather, it is the idea how to circumvent the weak exoelectrogenic activity of cyanobacteria by redirecting the electron flow. The design of a two-species microbial consortium guides the electrons flow from photons to an organic electron sink, then to electricity, with the aim to improve the energy conversion efficiency from light to electricity. We are pleased that we were able to demonstrate the feasibility of this novel idea. Scientifically, our work represents a new type of biophotovoltaic setup which utilizes the function of a defined

microbial consortium comprising a photoautotroph which stores the light energy into an organic compound and a heterotroph which converts the organic compound into electricity. Practically, this lactate-mediated consortium-type biophotovoltaic increased the power density by a magnitude compared with the mediatorless one, and the long life time achieved in this study increases our confidence on future application of biophotovoltaic.

It is not easy to make the two-species microbial consortium work in a constrained electron flow way. The questions that we encountered include: Whether we can create a compatible environment that can make both microbes happily work together? Whether the photoautotroph can produce sufficient amount of organic compound and whether the organic compound can be efficiently converted to electricity by the exoelectrogenic heterotroph? How long can the system be operated? Will the performance of such a system be better than the single species biophotovoltaic?

As pointed out by Reviewer #3, we performed manipulation at three levels to make the constrained electron flow happen in this two-species consortium. At the genetic level, we developed mutants that provide a significant out-flux of D-lactate (*S. elongatus*) and removal of an inhibitory enzyme (nitrate reductase) in *Shewanella*. At the environment level, we optimized the growth medium, oxygen, light, etc to make the two species work together. At the reactor level, we designed three setups and demonstrated one of the setups can maximize the potential of the two-species biophotovoltaic and achieved a longevous operation. Therefore, the idea may be simple, but the two species are not necessarily to work together without the aforementioned systematic design.

2) The authors have a made a number of strange choices in their experimental design. Although it is understandable that during the scientific discovery process, sometimes pragmatic choices need to made, for publication it is still imperative that experiments are then designed in a reductionist approach where only minimal changes are made to an experimental model to prove that the results are only due to these particular conditions. To give a couple of examples:

2a) *S. elongates* (*Syn2973*) was engineering to include the *omcS* gene of *Geobacter sulfurreducens*. I assume the authors have done this with the idea that *Syn2973-omcS* would be capable of extracellular electron transfer. However, this was not observed. The authors then continued to engineer

Syn2973 to express lactate using the *ldh* gene. However, rather than cloning this in *Syn2973*, *ldh* was cloned in *Syn2973-omcS* to create *Syn2973-omcS-ldh*. In order to prove that there are no unexpected results from the *omcS* gene, all experiments have to be repeated with a *Syn2973-ldh* strain using *Syn2973* as a negative control.

Response:

Thanks very much for your comment. In another study of our lab, the *omcS* gene was cloned into *Syn2973* to channel the photosynthetic electron into respiratory chain. The resulting strain *Syn2973-omcS* exhibited an increased intracellular NADH level, which would be in favor of production of the NADH-dependent chemicals including D-lactate (unpublished, data not shown). Therefore, when expressing the *ldh* gene in both strains *Syn2973* and *Syn2973-omcS*, the resulting strain *Syn2973-omcS-ldh* produces higher concentration of D-lactate than *Syn2973-ldh* does (supplementary Fig. 3), but the mechanism remains to be studied. Consequently, we selected the D-lactate high-producing strain *Syn2973-omcS-ldh* for microbial consortium construction, whereas strain *Syn2973-omcS* which does not produce D-lactate was used as a negative control. We have clarified this result in Line 135-141.

Yes, we are aware of the electron transfer ability of *omcS* gene and we also wonder whether the introduction of *omcS* would change the exoelectrogenic activity pattern of the host cyanobacterial strains. Determination of the exoelectrogenic activity of strains *Syn2973*, *Syn2973-omcS*, *Syn2973-ldh*, and *Syn2973-omcS-ldh* showed that all strains could generate detectable currents which were below $50 \text{ mA}\cdot\text{m}^{-2}$, but no significant differences were observed among these four strains (supplementary Fig. 9). This indicates that *omcS* gene did not contribute to the increase of the exoelectrogenic activity of the CLS consortium in the present setup. Strain *Syn2973-ldh* produces less D-lactate as compared to that of strain *Syn2973-omcS-ldh*. Thus, repeating the experiments with strains *Syn2973* and *Syn2973-ldh* would result in a lower power output, but would not change the conclusion of this study. We are more interested in testing strains with higher D-lactate-producing ability in further study. We therefore hope the editor and the reviewer would agree that the suggested experiment is not required for this work. The relevant description can be found in Line 198-201.

2b) In the “dark cell”, the cathode contains an electrode with 50 mM

ferricyanide. This is rather unusual. Why is the cathode not an oxygen reducing cathode? Currently, because only ferricyanide is used, the potential of the cathode is undefined and subject to fluctuations as some of the ferricyanide is converted ferrocyanide during the course of the experiment. As such, the potentials measured have little quantitative meaning. Similarly, can the authors experimentally confirm that the power output is not limited by the cathode (although I do not expect this to be case).

Response:

Thanks very much for your comment. Ferricyanide is a common cathodic electron acceptor due to its rapid reaction rate, good solubility and low overpotential. For this reason, ferricyanide is widely used in laboratory-scale basic research. According to the theoretical calculation (Eq. 1), 50 mM $K_3[Fe(CN)_6]$ is sufficient for running our device at maximum power density ($P_{max}=150 \text{ mW}\cdot\text{m}^{-2}$ and $I_{max}=350 \text{ mA}\cdot\text{m}^{-2}$) for at least one month.

$$t = \frac{Q}{I_{max}} = \frac{n \cdot m \cdot F}{I_{max}} = \frac{1 \times 50 \times 10^{-3} \text{ mol/L} \times 0.14 \text{ L} \times 96485 \text{ C/mol}}{350 \times 10^{-3} \text{ A/m}^2 \times 6.25 \times 10^{-4} \text{ m}^2} = 3.09 \times 10^6 \text{ s} = 35.7 \text{ d} \quad (1)$$

Where Q (C) is the maximum amount of Coulombs can be accepted when total ferricyanide was reduced to ferrocyanide, n is the number of electrons acquired when one molecule of ferricyanide was reduced ($n=1$), m (mol) is the molar amount of ferricyanide in one device, and F is the Faraday constant ($96485 \text{ C}\cdot\text{mol}^{-1}$).

We also experimentally confirmed that there was no cathodic limitation in our setup – increasing the concentration of ferricyanide over 50 mM has little effect on the current density, nor does enlarging the size of cathode (Supplementary Fig. 1, the relevant text can be found in Line 119-124). In the final setup for long-term operation, 200 mM $K_3[Fe(CN)_6]$ was added to ensure the adequate supply of electron acceptor. This was clarified in the Methods section (Line 562-564). The decrease of current density in the later period was mainly due to the gradual consumption of lactate and the decrease of the stability of the biofilm.

Oxygen is the most common electron acceptor used in the cathode compartment due to its high oxidation potential, low cost and the clean

products it yields. However, oxygen is not a perfect electron acceptor due to its poor contact with electrode and the slow kinetics of the oxygen reduction. Although the cathodic reaction can be improved by the use of catalytic-coated electrodes, the catalysts (like Pt) are quite expensive so there is a need to develop inexpensive catalysts. In our future experiments, we will consider developing oxygen cathode (especially air-cathode).

2c) The experiment in Figure 7 seems to have been performed only once. Strangely, however, the first 15 days, the culture media contained 5% LB, which was then omitted from day 15 onwards. This seems arbitrary, but more importantly, why was the experiment not repeated in the absence of LB, as LB is hypothesised by the authors to have a detrimental effect on the power output.

Response:

Thanks very much for your comment. In the setup with medium replenishment, *S. oneidensis* cells in the anodic chamber were pumped out along with the anodic culture. Therefore, 5% LB was added to the medium for regenerating *S. oneidensis* cells and maintaining its activity. However, the current density was neither very high nor stable during the period of 0–13th days. Moreover, we observed excess growth of *S. oneidensis*, which might result from consuming D-lactate, thus decreased the proportion of D-lactate used for current production. We postulated the high concentration of LB might be accounting for this, so we empirically decreased the addition of LB to 1% from the 13th day onward. Interestingly, we found this operation gradually restored the current production and maintained the current density at a high level stably. During the revision process, we performed additional experiments to verify this empirical change. We show it is essential to add LB to the medium, otherwise the *S. oneidensis* would be washed out; and the optimum concentration of LB for maintaining the cell density of *S. oneidensis* and achieving stable current output is 1% (Supplementary Fig. 12). We have clarified this issue and the relevant text can be found in Line 325-337.

2d) In the “dual flow cell” (Fig. 3c) *Syn2973* will be pumped from the first to the second compartment. The influence of creating a mixed culture in the second cell is not explored. Besides lactate, it is clear that transferring the culture (including *Syn2973*) will transfer a wide variety of metabolites to the second culture. What is the effect of doing this?

Response:

Thanks very much for your comment. In the flow setup (Fig. 2c in the revised manuscript), the cyanobacteria culture (left) was pumped into the anodic chamber (right). In the anodic chamber, two cultures were mixed and *S. oneidensis* cells use the metabolites of cyanobacteria for current production. The process of mixing was the same with that in the temporal separation setup (Fig. 2a in the revised manuscript), so we don't think we need to specifically describe the creation of the mixed cultures in the flow setup.

In the temporal separation setup, we found the mixture of two strains produced more current than the mono-culture of *S. oneidensis* supplemented with equal amount of D-lactate (Fig. 4a). This indicates, as pointed out by the reviewer, additional chemicals were produced by strain *Syn2973-omcS-ldh*, and these chemicals were used by *S. oneidensis*. On one hand, we analyzed the HPLC spectrum of the extracellular metabolites of strain *Syn2973-omcS-ldh*. Besides D-lactate which is the dominant secreted metabolite in *Syn2973-omcS-ldh*, we also found 20-30 mg/L formic acid was produced (Supplementary Fig. 8c). As formic acid is also a substrate of *S. oneidensis* for current production, this could partially account for the phenomenon we observed. On the other hand, cyanobacteria may release their endogenously stored compounds through cell lysis and dark fermentation during the discharging process, which may also serve as substrates for *Shewanella*. We have supplemented relevant description in Line 234-245 for clarification.

3) The manuscript needs to be considerably shortened. Long sections are devoted to minor issues such as optimisation of media conditions. Although these 'small' issues are important, the results of such optimisations can be reported in a couple of sentences.

Response:

Thanks very much for your constructive comment. We have merged the part 2 and part 3 in the Results section, and the length was significantly shortened. Please refer to Line 143-186.

4) The way the results are reported is confusing at points. For instance, most experiments only report the potential of the electrochemical cell. It is not till the methods section that it is mentioned a 2.0 kOhm resistor was used between

anode/cathode. As mentioned above, it would have been preferable if the current was measured too, to make sure that internal resistances in the anode/cathode do not further limit the current.

Response:

Thanks very much for your constructive comment. We have revised all figures and the relevant results were shown as current density. We described the details of current density calculation in the Methods section (Line 512-516).

Reviewer #2 (Remarks to the Author):

In this study the authors have presented a very interesting novel bio electrochemical setup and results. Based on the current issue of having the power outputs of the mediatorless BPV systems demonstrated not exceeded few $\text{mW}\cdot\text{m}^{-2}$, the authors has proposed and test the formation of a two-species consortium (*Synechococcus elongatus* UTEX 2973 + *Shewanella*). This was implemented via three experimental setups with temporal and spatial separation as shown in figure 3 of this manuscript.

To the best of my knowledge, the findings presented in this study are novel and will be well perceived by the academic community in the field of bio energy. As shown in figure 7, maintaining the power output over 50mW m^{-2} for many days is a quite remarkable result.

Response:

Thanks very much for your positive comments.

Before publication the authors should, in my view, address the following points:

-1-

The word “efficient” is used in the title and several parts of this manuscript.

I find this a bit confusing as “efficient” is generally used to define the conversion rate of light into electricity. In this study this term is used to refer to the energy conversion efficiency of D-lactate to electricity. This might be a legitim choice taken by the authors, but it needs to be explained in the title. Otherwise the title could be misleading.

Response:

Thanks very much for your comment. You are right, that the energy conversion

process from D-lactate to electricity can be termed as “efficient”. Although the power output of our system was one order of magnitude higher than that of the previously demonstrated mediator-less BPV systems, the overall energy efficiency from light to electricity is much lower than that those of the PV systems. So we revised the title as “Development of a longevous two-species biophotovoltaics with constrained electron flow”.

-2-

In figure 4d, is the value represented by the blue bar (OD 0.1) significantly bigger than the values represented by the grey bars (ODs 0.01, 0.05, 0.5, 1.0 and 2.0)?

Please comment on this point.

Response:

Thanks very much for your comment. There was no significant difference among these bars. The blue bar (OD 0.1) in Fig. 3d (the previous Fig. 3d) just indicated we designated OD_{600} of 0.1 as the optimal inoculation density in the following experiments. The blue color was changed to the same color of the others.

-3-

It is not clear to me how the calculation of current/power output are performed. At the line 576 - 577 the authors have declared that “The current and power were normalised to the projected geometric area of the anode (6.25 cm^2)”. This is perfectly fine if 6.25 cm^2 is the total active surface area of the experimental setup used in this study. For total active surface area I mean the area exposed to light. In other words, as the presented device is powered by light, therefore the declared current/power output needs to be normalised to this surface area. Please comment or correct.

Response:

Thank you very much for your comment. BPV systems developed from different research groups were characterized under a wide variety of conditions and configurations. This includes the status of the microbes (in biofilm or in suspension), the spectra of illumination, the geometry and size of the devices, the type of electrodes, et al. Therefore, the standardized characterization conditions have yet to be agreed and adopted by the community. Even under such circumstances, almost all studies used the

current/power outputs normalized to the active surface area of the anode to compare the performance of devices with different configurations and different operation conditions (refer to citation 4, 27, 34). Taking this into consideration, the calculation of current/power density was also normalized to the active surface area of the anode (6.25 cm²) in this work. The details of the calculation method can be found in Line 512-516.

Moreover, we displayed the results as current density for relevant figures in revised manuscript.

-4-

Fig. 7

It is not clear to me what happened at day 14 to when the performance of the device increased sharply. The concentration of lactate appears to grow gradually from ca. 2 mM (day 2) to ca. 3.5 mM at day 12, then it appears to stay stable around 3.5 mM from day 12 onward.

The reduction of fresh LB from 5% to 1% per day should not be the cause for this variation.

Please comment or correct.

Response:

Thank you very much for your comment. The increase of the performance of the device could be ascribed to the increased and stable production of D-lactate. On the other hand, reduction of LB from 5% to 1% was also a key factor. Please refer to our response to question 2c raised by reviewer #1.

-5-

Please add an additional diagram showing the actual geometry and size of your experimental setup to complete the information given in figure 1, figure 4 and sup figure 7.

Response:

Added as suggested, please refer to Supplementary Fig. 14a.

-6-

In supplementary figure 6 it is said that Potassium ferricyanide inhibited cyanobacteria growth and finally killed the cells even at a low concentration of 1 mM under light.

Whether I do not intend to dispute the finding here reported I need to point out that in a study actually cited in this manuscript (citation 7) it is reported that ferricyanide at concentration up to 30 mM within 30 hours experiment run is not compromising the rate of oxygen evolution in *Synechocystis* 6803 wt (Sup figure 4).

Can the authors comment on this point?

Response:

Thank you very much for your comment. The key finding of this study is to confirm that ferricyanide will be decomposed to $\text{Fe}(\text{OH})_3$ and KCN once exposed to light (ref 38). Ferricyanide itself may not be harmful, but its decomposed product KCN is harmful for cyanobacteria.

In our experiment, we mixed cyanobacteria with different concentration of ferricyanide and the mixtures were incubated under full spectrum fluorescent lamps, which is more energetic due to blue band and ultraviolet region are contained. In addition, the mixtures were subject to shaking at 140 rpm during the whole process. These experimental conditions are in favors of accelerating ferricyanide decomposition.

It was not clear from reference 9 (the previous reference 7) whether the mixtures of cyanobacteria with different concentration of ferricyanide was incubated in the light or in the dark before the measurements of oxygen evolution and power output. If it was conducted in the dark, the ferricyanide would not be decomposed to KCN thus would not show the toxicity. If it was conducted in the light, the decomposition of ferricyanide would be slowed down greatly due to the light source is an array of red LEDs (emission peak at 625 nm) and the mixtures was incubated at a static state.

Therefore, to our opinion, our experiments and the experiments in reference 9 were performed at different conditions. We have clarified these differences in Line 384-393.

-7-

It would be very informative to have a graph showing the electrical outputs (e.g. current) versus concentration of lactate.

Response:

Thank you very much for your suggestion. We have added a graph showing the maximum current density versus the concentration of D-lactate (Supplementary Fig. 2c). The relevant description can be found in Line 132-135.

Reviewer #3 (Remarks to the Author):

The manuscript submitted by Li, Zhang and co-workers describes their design and characterization of biophotovoltaic cells that are operated by the cooperation of two bacterial species – a photosynthetic cyanobacterium (*S. elongatus*) and *Shewanella*. The premise of the study is that direct solar energy conversion by photosynthetic organisms to such cells is not as efficient as obtained from exoelectrogenic *Shewanella*, while the latter bacteria require a constant influx of oxidizable nutrients. The two organisms are not necessarily compatible in their typical growth media, and so the group performed a series of optimization experiments to try to find the optimal conditions for both species, with the resulting power output (magnitude and time) serving as the optimizing parameter.

Optimization required manipulation of the system on all levels. Both biological components were mutants that provide a significant out-flux of D-lactate (*S. elongatus*) and removal of an inhibitory enzyme in *Shewanella* (nitrate reductase). Other parameters were growth media, temperature, light, etc.). Following the first step in the design (choosing the correct cyanobacterial mutant), the authors describe at some length the optimization of the media to prevent inhibition and/or support growth. While perhaps interesting, this section is long – and in the end, utilizing both bacteria in the same cell was not the best solution. This section could be significantly shortened.

Response:

Thank you very much for your constructive comments. We have merged the part 2 and part 3 in the Results section, and the length was significantly shortened. Please refer to Line 143-186.

The results of the study are quite impressive, and show potential for device that can provide extended solar energy conversion. The final device separates between the cyanobacteria and *Shewanella* and provides a continuous flow of

nutrients and buffer components that result in continued viability of the cultures of both organisms over 40 days. The authors provide an extensive table of many of the comparable Bio-photovoltaic (BPV) systems developed and published over the past few years. Two studies of similar nature published in this journal were omitted from this list: Pinhassi et al. 2016 and Saper et al. 2018. In the former, spinach membranes provided maximal power density of 2500 mA/m² (far greater than the power obtained here) but for far shorter time than presented here. In the latter, live cyanobacterial cells overlaid onto a graphite electrode provided less power, but continued to provide current when glucose was added to the buffer (and so is similar to the lactate providing the driving force for the *Shewanella* in the present study). In both of these studies, hydrogen gas was produced at the cathode. Hydrogen has an advantage as a potential fuel, however more importantly, in this fashion the concentration of the species reduced at the cathode remains constant. The reaction at the cathode in the present work is rather ignored. Over the course of 40 days of current transfer, one would expect the concentration of ferricyanide to significantly decrease, thus changing the redox potential of the cathodic chamber, and the resulting difference in potential with the anodic chamber. This issue must be clarified before acceptance.

Response:

Thank you very much for your positive and constructive comments. These are two excellent and very important works in BPV field, which helped us to better understand the mechanisms of photocurrent production in live cyanobacteria and thylakoids. Moreover, in these two studies, BPV systems were not only used for current production but also developed to produce hydrogen fuels, which expands the application of BPV technology. These two studies were cited and listed in Supplementary Table 2.

In these two studies, H⁺ was used as electron acceptor and hydrogen evolution occurs at the cathode. However, an applied bias (applied potential) needs to be provided for hydrogen evolution, which means these devices require an external power source to operate. We operate our devices without providing the applied bias. Thus, the hydrogen evolution cathode is not suitable for the present operation.

With regard to the question whether the redox potential of the cathodic chamber would be changed along with the ferricyanide was reduced, we have

experimentally demonstrated 50 mM ferricyanide is sufficient in our setup, and the increase or decrease of ferricyanide concentration did not significantly influence the current density. This indicated there was no significant change in redox potential of the cathodic chamber during the process (Supplementary Fig. 1, Line 119-124). Please refer to our response to question 2b raised by Reviewer #1 for more details.

Specific comments:

1. The results section starts with the description of different stages of optimization – for instance D- vs. L- lactate, using the voltage produced as the parameter to be maximized. However, the cell itself has not yet been presented and it is not clear to what the obtained voltage has been measured/calibrated. The explanation that is given later in the manuscript (in the methods section, is also not clear enough.

Response:

Thank you for your comments. A brief introduction on the device used in this study and current production measurement was added at the beginning of the Results section (Line 116-119). The methods of measuring the voltage and calculating the current/power density were clarified in the Methods section (Line 512-516).

Besides, current density, rather than voltage, was displayed in the relevant figures in the revised manuscript.

2. The *S. elongatus* mutant produces and exports D-lactate. What is the effect on cell growth? How much of solar energy is converted to lactate production, versus other cellular requirements? Perhaps this could be added to SI Fig. 4? In addition, what was the effect of the Δ napA mutation on *Shewanella* growth?

Response:

Thank you for your constructive comments. We have added one figure showing the cell growth of the cyanobacteria strains (Supplementary Fig. 8a). As we can see, the D-lactate-producing strain grew better in MBG11 than the control strain did. This could be due to that production of D-lactate consumed additional reducing equivalent thus helped enhancing the photosynthesis. This result was present in Line 192-195.

We have added a figure showing the cell growth of two *Shewanella* strains (Supplementary Fig. 6). The deletion of *napA* has no significant effect on cell growth (Line 163-164).

The solar energy partitioning in D-lactate and other cellular requirements (mainly biomass) in *Syn2973-omcS-ldh* was calculated based on the heat of combustion. The calculated results were showed in Fig. 7. The relevant description can be found in Line 427-429. The calculation method was clarified in the Methods section (Line 590-608).

3. SI Fig. 2 panels should be explained in some detail: B - top drops vs. bottom, serial dilution? C- reference for correlation between fluorescence and ROS; D- method of measurement of ORP? Mention in the text of what is included in the Materials and Method section would also be helpful.

Response:

Thank you very much for your suggestion. SI Fig. 7 in the revised version was explained in more details in its legend and the Supplementary Methods section. B- from left to right: 10-fold serial dilution. C- the references have been added (Supplementary file, Line 40-41). D- a simple handheld ORP Meter was used for measurement of ORP by immersing the probe into the cell culture and the result was displayed directly. This method has been described in the legend and the Supplementary Methods section (Supplementary file, Line 26-30).

4. The authors quote the level of activity of a certain architecture based on the voltage obtained between anodic and cathodic cells (in mV). As the carbon cloth electrode must actually come into contact with the *Shewanella* cells, is the source of the elevated (and then decreasing) voltage due to the state of the bacterial film's connection with the anode? Since the total size of the cell is large, filled with bacteria and receiving the lactate from the cyanobacteria, why not make the anode much bigger? These issues should be explained in more detail.

Response:

Thank you for your constructive comments. You are right, the status of biofilm in the anode influence the current density. The current density went up along with the formation of biofilm at the beginning, but decreased thereafter when

the biofilm gradually became unhealthy, associated with the gradual consumption of lactate.

We have performed an additional experiment of enlarging the size of anode from 2.5 × 2.5 cm to 5.0 × 5.0 cm, which is the maximum available occupancy of our device. The results showed the current output improved, whereas the current density decreased, along with the enlargement of anode (Supplementary Fig. 10a, b). Moreover, the maximum power density is inversely proportional to the anode area in this size range (Supplementary Fig. 10c). These results indicated the smaller the size of the anode, the higher the current density and power density it produced. These results were included in the Results section (Line 267-273).

5. The elevated power output associated with the consortia of bacteria was suggested to be due to some electrogenicity on the part of the cyanobacteria. This is indeed one possibility, however is it not also possible that there is a certain proportion of the cyanobacterial cells that die, lyse and “donate” their internal storage molecules which can be utilized by the *Shewanella*?

Response:

Thank you for your comment. Yes such possibility does exist. We have addressed this possibility in the revised manuscript (Line 242-245). Cyanobacteria may release their endogenous stored molecules through cell lysis or dark fermentation during the discharging process, which may provide more substrates for *Shewanella*.

Reviewers' comments:

Reviewer #1 (Remarks to the Author):

Complex photo-microbial fuel cells (MFCs) comprises a wide variety of bioelectrochemical systems where autotrophic species are mixed with heterotrophic species, where the heterotrophic species are typically exoelectrogenic and used to convert organic carbon produced by the autotrophic species into electricity. For recent reviews we refer to a review by McCormick et al, 2015 (DOI: 10.1039/C4EE03875D), who also cite earlier reviews in this area.

In this paper by Zhu et al., a complex photo-MFC is tested based on previously developed cyanobacteria (the autotroph) that produces lactate (the organic carbon) that is fed to a *Shewanella oneidensis* (the heterotroph and exoelectrogen). The paper describes in detail an extensive set of optimizations in growth media and the cell setup, after which a system is reported that produces $> 10 \mu\text{W}/\text{cm}^2$ for over 40 days, which sets a new benchmark in the field.

In this revised version, the authors have addressed all technical comments I made.

Response:

Thank you very much for your positive comments. The review mentioned by the reviewer is a milestone reference in this field and it was also cited in our manuscript.

Reviewer #2 (Remarks to the Author):

The improvements provided by the authors are, in my view, sufficient for permitting publication. I would like to stress that, reporting a photosynthetic bio-electrochemical system (BPV system) where the power output is maintained over $50 \text{ mW}\cdot\text{m}^{-2}$ for many days is a quite remarkable result. Longevity associated with reasonable high power output for biological systems are key issues. Those are probably the key results of this study which justifies publication in Nature Communications.

Response:

Thank you very much for your positive comments.

Reviewer #3 (Remarks to the Author):

The manuscript resubmitted by Li, Zhang and co-workers describes their design and characterization of biophotocatalytic cells that are operated by the cooperation of two bacterial species – a photosynthetic cyanobacterium (*S. elongatus*) and *Shewanella*. As described in the original manuscript the premise of the study is that direct solar energy conversion by photosynthetic organisms in such cells is not as efficient as obtained from exoelectrogenic *Shewanella*, while the later require a constant influx of oxidizable nutrients.

The resubmitted manuscript is greatly improved. The authors have answered most of the comments and questions posed by the reviewers, including my own.

Response:

Thanks very much for your positive comments.

However two aspects of the study are still troublesome and require further explanations prior to acceptance.

1. In Fig. 4c, a polarization curve is presented. This shows that there is a change in the current density as a function of potential. I am assuming that the potential is measured between the two electrodes (anodic and cathodic) relative to the reference electrode - although this is not actually described in the Methods section. Wouldn't increasing the potential between the electrodes assist in the reduction of ferricyanide on the cathode, thereby increasing the current density? Why is there an increase in current density at low potentials – up to an outstanding 1.4 A/m^2 ?

Response:

Thanks for your comments and questions. The polarization curves were obtained by performing linear sweep voltammetry (LSV) analysis. LSV analysis was conducted on a two-electrode configuration (the working

electrode was connected to the anode, whereas both the counter electrode and the reference electrode were connected to the cathode). The applied potential on working electrode was changed from the open circuit potential (OCP, $\varphi_{\text{anode}} - \varphi_{\text{cathode}} = -0.88$ V) to zero V, as typically used for the characterization of MFC. In this operation, the applied potential equals to the potential difference between the anode and cathode. When the applied potential equals OCP, the MFC is in the open circuit state, the current density is zero; when the applied potential equals zero V, the MFC is in the short circuit state, the current density is the maximum. The details about how to perform LSV analysis have been further clarified in the Methods section (Line 593-600).

The measurement of a two-electrode configuration represents the overall performance of an electrochemical cell. In this study, the overall performance of BPV system, rather than anode performance, was evaluated by using two-electrode LSV analysis. Therefore, a non-polarizable reference electrode, such as Ag/AgCl reference electrode, was not introduced. In the polarization curve, the Y-axis represents the absolute value of the potential difference between the anode and the reference electrode. As the reference electrode was connected to the cathode, the Y-axis also represents the potential difference between the cathode and the anode (that is the voltage produced by BPV system). Nonetheless, since the working electrode was connected to the anode, increasing the potential between the anode and the cathode would only change a little to the cathode potential, thus would not assist in the reduction of ferricyanide on the cathode too much.

As described in Logan, B. E. *et al.*, 2006 (DOI: 10.1021/es0605016, citation 27), the typical polarization curves can be generally divided into three zones:

- (i) Starting from the OCP at zero current, there is an initial steep decrease of the voltage. The activation losses are dominant in this zone.
- (ii) The voltage then falls more slowly and the voltage drop is fairly linear with the current. The ohmic losses are dominant in this zone.
- (iii) There is a rapid fall of the voltage at higher currents. The concentration losses (diffusion limitation) are dominant in this zone.

For the polarization curve of the CLS microbial consortium shown in Fig. 4c, we did not observe the rapid fall of the voltage at higher currents. Instead, the current density showed a significant increase when the voltage was close to

zero V. This phenomenon can be explained from two aspects:

(1) In the CLS microbial consortium, the electron donor, D-lactate, was adequate. The cyanobacteria surrounding the biofilm of *S. oneidensis* could also produce more D-lactate to be used for current production. Thus, we argue that the diffusion limitation of electron donor might not exist in the CLS microbial consortium even at low voltages (that is at higher currents). Thus, no rapid fall of the voltage at higher currents was observed.

(2) In a bioelectrochemical system, the anode is the terminal electron acceptor for microorganisms. In order for the reaction of transferring electrons outside to the anode to be thermodynamically favorable, the anode must have a higher (more positive) potential. Besides, a recent study (Hirose, A. *et al*, 2018. *Nat. Commun.*, 9, 1083, citation 29) found that high anode potentials facilitated current generation of *S. oneidensis*, which was resulted from upregulated D-lactate oxidation and NADH oxidation at higher anode potentials.

Thus, on the premise of adequate electron donors in the CLS microbial consortium, the oxidation of electron donors would be accelerated and more electrons would be transferred to the anode when the anode potential (versus the cathode) increased from -0.88 V to 0 V. This would lead to a significant increase on current density when the potential was close to zero V. Therefore, we speculated the electron transfer process in the CLS microbial consortium would change from the typical diffusional controlled mode to the kinetic controlled mode at low voltages, thus explaining why there was a significant increase on current density at low voltages. The relevant discussion was supplemented in the text (Line 259-270).

2. I agree with the two other reviewers that the results presented in new Fig. 6 are not clear. The explanations given by the authors may contribute, however the sharp changes in power/current output are not really explained in the rebuttal or the text. This needs to be further explained. Also, the response claims that additional experiments were performed, but the results of these new experiments are not presented nor are error bars added to the graph.

Response:

Thanks for your comments. The observed “sharp” change in current output from the 13th day onward was actually not a rapid change – this process

lasted for 7 days. There were two contributing factors: one is that the LB addition was decreased to 1%, and another is that more D-lactate was produced. Adding a suitable amount of LB is crucial for maintaining the cell density in this flow-setup and our additional experiments also verified this statement. The supplementary data showed it is necessary to add LB to the medium, otherwise the cells of *S. oneidensis* would be washed out; and the optimum concentration of LB for maintaining the cell density of *S. oneidensis* and achieving stable current output is 1% (Supplementary Fig. 12, shown below).

Supplementary Fig. 12 | Current production in mono-culture of strain *S. oneidensis-ΔnapA* with medium replenishment. The MBG11 media containing 3.5 mM D-lactate with different LB additions (0%, 1%, 3% and 5%) were used to replenish the anodic electrolytes. (a) The current density produced in replenished setups. (b) The cell density of *S. oneidensis-ΔnapA* in anodic chambers during the whole process.

Supplementary Fig. 12 also shows when LB addition was 5%, the current density first increased then started to continuously decline. When LB addition was 1%, the current density continuously increased. This feature can help us to better understand the continuous increase of current density during the 13-20th days in Fig. 6 when the LB addition was decreased from 5% to 1%.

It is likely that when LB addition decreased from 5% to 1%, the steady state of biofilm of *S. oneidensis* was disturbed and the biofilm needed to be reconstructed so as to reach a newly established steady state. This process lasted for 7 days, and the profile is similar to that of the process at the initial stage (Fig. 6, 0-3rd days), during which the current density increased

continuously along with the gradual formation of the biofilm of *S. oneidensis*.

Moreover, in the first half day after switching to 1% LB (Fig. 6, 13-13.5th day), the current density actually decreased a little (rather than immediately increased). This decrease is then followed by a steady increase on current density over the next 6.5 days (during the 13.5-20th days). In fact, there were many current density increase/decrease oscillations during the steady state, which is an indication of the partial disruption/partial reconstruction of the biofilm of *S. oneidensis*.

Therefore, combining the data shown in Fig. 6 and Supplementary Fig. 12, we are convinced that the steady increase of current density during 13-20th days is mostly correlated with the decreased addition of LB. We have further clarified the relevant description (Line 341-360).

Furthermore, current density was recorded every 20 minutes for this longevity testing experiment. This means a 40-day experiment would yield $40 \text{ days} \times 24 \text{ hours/day} \times 3 \text{ data points/hour} = 2880 \text{ data points}$. So we used smooth curve which connects all the data points to show the profile of the current density. If we display the error bars of each data points, the figure would be a mess and it will be very difficult to interpret the data shown in the figure. Therefore we showed the data from one of the experiments to demonstrate we are able to run a longevous two-species biophotovoltaics.

We have supplemented additional descriptions in the legend of Fig. 6 to clarify this: "Current density was recorded every 20 minutes. The data shown in the different curves were from one of the experiments, each contains more than 2800 data points."

Reviewer' comments:

Reviewer #3 (Remarks to the Author):

The explanations provided by the authors to my queries and the changes to the text are sufficient to support publication.

Response:

Thank you very much for your positive comments.